# Head and Neck Cancer and Sarcopenia: An Integrative Clinical and Functional Review

**DOI:** 10.3390/cancers16203460

**Published:** 2024-10-12

**Authors:** Kazuhira Endo, Mariko Ichinose, Eiji Kobayashi, Takayoshi Ueno, Nobuyuki Hirai, Yosuke Nakanishi, Satoru Kondo, Tomokazu Yoshizaki

**Affiliations:** Division of Otolaryngology, Head & Neck Surgery, Graduate School of Medical Science, Kanazawa University, Kanazawa 920-8640, Japan; m_ichinose@med.kanazawa-u.ac.jp (M.I.); e_kobayashi@med.kanazawa-u.ac.jp (E.K.); uenotaka@med.kanazawa-u.ac.jp (T.U.); nhhira@med.kanazawa-u.ac.jp (N.H.); nakanish@med.kanazawa-u.ac.jp (Y.N.); ksatoru@med.kanazawa-u.ac.jp (S.K.); tomoy@med.kanazawa-u.ac.jp (T.Y.)

**Keywords:** head and neck squamous cell carcinomas, sarcopenia, chronic inflammation, CT imaging

## Abstract

**Simple Summary:**

Head and neck squamous cell carcinoma (HNC) is a heterogeneous group of malignancies that originate in the upper aerodigestive tract. Sarcopenia, which is a condition characterized by the loss of skeletal muscle mass and function, is a prevalent condition among cancer patients, significantly impacting their overall health and treatment outcomes. This paper provides a comprehensive review of the relationship between HNC and sarcopenia, focusing on the underlying mechanisms, clinical implications, assessment methods, and potential interventions. The aim is to enhance our understanding of the complex interplay between these two conditions and guide the development of personalized therapeutic strategies to improve patient outcomes.

**Abstract:**

Sarcopenia is recognized as a crucial factor impacting the prognosis, treatment responses, and quality of life of HNC patients. This review discusses various mechanisms, including common etiological factors, such as aging, chronic inflammation, and metabolic dysregulation. Cancer-related factors, including tumor locations and treatment modalities, contribute to the development of sarcopenia. The clinical implications of sarcopenia in HNC patients extend beyond reduced muscle strength; it affects overall mobility, reduces quality of life, and increases the risk of falls and fractures. Sarcopenia serves as an independent predictor of postoperative complications, chemotherapy dose-limiting toxicity, and treatment outcomes, which affect therapy planning and perioperative management decisions. Methods to assess sarcopenia in HNC patients encompass various techniques. A sarcopenia assessment offers a potentially efficient and readily available tool for clinical practice. Interventions and management strategies for sarcopenia involve exercise interventions as a cornerstone; however, challenges arise due to patient-specific limitations during cancer treatment. A routine body composition analysis is proposed as a valuable addition to HNC patient management, with ongoing research required to refine preoperative exercise and nutrition programs for improved treatment outcomes and survival.

## 1. Introduction

Head and neck cancers (HNCs) are the ninth most common malignancies worldwide. They encompass a diverse group of malignancies originating from various anatomical sites, including the oral cavity, pharynx, larynx, and paranasal sinuses. Despite significant advances in treatment modalities, HNCs remain a substantial public health concern due to high morbidity and mortality rates [1]. Sarcopenia, characterized by the progressive loss of skeletal muscle mass (SMM) and function, is associated with various negative clinical outcomes across multiple cancers [2,3]. Notably, patients with HNC face a significantly higher risk of malnutrition than those with other malignancies [4]. Sarcopenia is emerging as a critical factor affecting the prognosis, treatment responses, and quality of life (QOL) of HNC patients. Patients with malignancies universally exhibit a low muscle mass, with a prevalence from 11 to 74% within this patient cohort that depends on the type and stage of cancer [5,6,7,8]. The prevalence of sarcopenia in HNC ranges between 35.5 and 54.5% [9]. Inconsistencies in definitions and cut-off values for sarcopenia across studies have contributed to variability in prevalence data [10].

## 2. Mechanisms Underlying This Relationship

The relationship between HNC and sarcopenia is complex and multifactorial (Figure 1). Common etiological factors, such as aging, chronic inflammation, and metabolic dysregulation, contribute to both conditions. The risk of sarcopenia is higher in HNC than in other cancers and has been associated with a reduced oral intake due to disease-related limitations, leading to malnutrition. The presence of large tumors may obstruct the aerodigestive system, worsening malnutrition. Factors including tumor locations and stages contribute to dysphagia, further impacting the dietary intake and nutritional status of HNC patients [9]. This, in turn, plays a pivotal role in the development and progression of sarcopenia in HNC patients [11]. Furthermore, treatments for HNC, such as surgery, radiation therapy, and chemotherapy, significantly impact muscle mass and overall physical function. Surgery may involve the removal of both the tumor and muscle or other structures, disrupting normal chewing and swallowing physiologies. Radiation therapy and chemotherapy also cause fatigue and decrease appetite, which further exacerbate muscle loss and contribute to swallowing dysfunction. Therefore, a large percentage of patients become dependent on feeding tubes during chemoradiotherapy [12]. Previous studies reported that between 3 and 52% of HNC patients developed anorexia and malnutrition during chemoradiotherapy [4,13,14]. Conversely, malnutrition contributes to the loss of swallowing muscle mass and decreased muscle strength, which worsen sarcopenia [15,16]. Cisplatin, a standard chemotherapy for HNC, has several significant adverse effects, including appetite suppression and decreased taste sensation [17].

In recent years, immune checkpoint inhibitor (ICI) therapy has emerged as a promising therapeutic strategy, attracting considerable attention and significantly enhancing the survival rates of some cancer patients, including those with HNC. Intensive research has focused on the relationship between body composition and ICIs [18]. This is driven by the effectiveness of ICIs being intricately linked to the host’s immune system, which is, in turn, significantly affected by body composition. Sarcopenia adversely impacts the effectiveness of ICIs in patients diagnosed with various malignancies; however, the specific mechanisms underlying this relationship remain unclear [19]. Friedlander et al. revealed a correlation between the extent of malnutrition, tumor burden, and overall outcomes. Additionally, the observed immune suppression in malnourished individuals has been linked to unhindered tumor growth [20]. Therefore, it is important to offer robust nutritional support and enhance patient education and support to ensure treatment continuity [21].

Sarcopenia is characterized as an age-related and progressive disorder, suggesting a gradual and natural development. Cancer-related sarcopenia, which is affected by factors beyond aging, may occur in conjunction with systemic diseases, involving inflammatory responses (e.g., malignancy), neurological disorders, or conditions such as osteoarthritis [22]. Previous findings support a relationship between tumor-induced systemic inflammation and the progression of sarcopenia, particularly through proinflammatory cytokines, such as interleukin-6 (IL-6) and tumor necrosis factor-alpha (TNF-α), which contribute to anorexia, skeletal muscle proteolysis, metabolic disturbances, and cancer cachexia [23,24,25]. Inflammatory signals reduce appetite, which aid in tissue repair and infection defenses. However, cancer hijacks these mechanisms, leading to uncontrolled catabolism, emaciation, and potential death. Tumors trigger muscle wasting by sending unidentified signals to muscle cells, activating protein degradation. Muscle atrophy may also be attributed to factors such as denervation or motor neuron diseases. Sartori and colleagues focused on the BMP pathway, which plays a crucial role in muscle function and mass regulation by inducing changes in gene expression through SMAD proteins [26]. Bilgic et al. revealed that tumor-induced oncostatin M up-regulated muscle EDA2R expression, which promoted muscle atrophy [27]. Excessive cytokine production may degrade skeletal muscle fibers, reducing their diameter and protein content, leading to metabolic breakdown, muscle proteolysis, and myocyte apoptosis [28]. Systemic inflammation and proinflammatory cytokines have been implicated in the development of sarcopenia. Low-grade inflammation, characterized by various cytokines and unique mechanisms, has been shown to significantly contribute to the progression of sarcopenia and its associated comorbidities, including insulin resistance, which leads to diabetes [29]. Skeletal muscle, far from being just an indicator of physical condition, plays a pivotal role as a multifaceted endocrine organ [30]. In this capacity, it actively releases a number of specific cytokines, commonly known as myokines, which participate in the intricate interplay between muscle physiology and overall systemic health. Skeletal muscle assumes a crucial role in regulating overall metabolic health, furnishing substrates for the energy requirements of other tissues. It supplies glucogenic amino acids, which are essential for liver gluconeogenesis, during fasting. An insufficient SMM may decrease the availability of these vital proteins. Moreover, as the largest insulin-sensitive organ, skeletal muscle plays a critical role in regulating the post-meal glucose and lipid balance [31].

However, clinical trials that evaluated the efficacy of anti-TNF therapies failed to prevent muscle atrophy in patients with advanced cancer cachexia [32]. While therapies targeting IL-1 and IL-6 have shown promise in patients with cachectic cancer, they have not consistently demonstrated a significant impact on SMM [33,34].

Some inflammatory markers are used as prognostic factors of malignancies, including the CRP-to-albumin ratio, neutrophil-to-lymphocyte ratio (NLR), and platelet-to-lymphocyte ratio (PLR) [35,36,37,38]. The correlation between sarcopenia and two systemic inflammatory markers, NLR and PLR, has been proposed as an indicator of skeletal muscle health. Given the cost-effectiveness and accessibility of these markers through routine blood tests, monitoring NLR and PLR regularly may serve as an effective approach for screening and managing sarcopenia [39].

## 3. Clinical Implications

Sarcopenia decreases muscle strength and physical function, resulting in difficulties performing daily activities [40]. Diminished mobility reduces QOL and increases the risk of falls and fractures. This heightened risk is attributed to skeletal muscle serving as a pivotal control center for systemic metabolic health [41]. Sarcopenia is closely linked to frailty, a state of increased vulnerability to stressors due to reduced physical reserves [42]. The assessment of SMM represents a valuable tool for identifying frail patients. Frailty, associated with unfavorable outcomes, is typically diagnosed through a time-consuming comprehensive geriatric assessment (CGA). Low SMM was independently linked to frailty, as assessed by the frailty screening G8 questionnaire, in 112 HNC patients [9]. Frailty screening questionnaires were utilized to identify candidates for CGA in 150 HNC patients (≥60 years old), and revealed a correlation between the G8 frailty score and SMM, particularly when not combined with handgrip strength (HGS) [43]. These findings suggest that sarcopenia effectively predicts frailty, offering a potentially efficient and readily available tool for clinical practice to select suitable patients for therapy.

Sarcopenia has significant clinical implications in patients with malignancies. It has been associated with increased postoperative complications, including pharyngocutaneous fistulas and wound complications, and longer hospital stays after surgery [15,44,45]. In a meta-analysis, low SMM was linked to severe postoperative complications in HNC patients [46]. Another study on elderly HNC patients revealed that low SMM correlated with early complications and readmission after surgery [47]. A prospective cohort study on 251 patients undergoing major head and neck surgery showed that sarcopenia (low SMM with low muscle strength or performance) was predictive of medical complications, higher complication grades, longer hospital stays, and poorer overall survival (OS) [48]. Similar findings were reported for patients undergoing free flap reconstruction after HNC resection. Low SMM was also associated with blood transfusion requirements in HNC patients undergoing free flap reconstruction. In some cases, low SMM was independently associated with discharge to post-acute care facilities [49]. Overall, low SMM and sarcopenia predict complications in major head and neck surgery, enabling the identification of high-risk patients for alternative treatment planning and perioperative management.

Sarcopenia also increases the rate of chemotherapy dose-limiting toxicity (CDLT), treatment delays, dose reductions, and failure to complete treatment in patients with a number of malignancies, such as lung cancer, renal cell cancer, colorectal cancer, breast cancer, and HNC [50,51,52,53,54,55,56]. Low SMM has been identified as a predictive factor for CDLT in studies on patients with locally advanced HNC receiving high-dose cisplatin-based chemoradiotherapy [57]. Patients with low SMM frequently developed CDLT, which led to dose reductions, treatment delays, or the discontinuation of chemotherapy [58]. Many studies consistently found a higher incidence of cisplatin DLT in patients with low SMM, and OS was significantly shorter in patients with than in those without DLT [59]. Additionally, low SMM was independently associated with DLT in multivariate analyses across different studies, emphasizing its significance as a predictive factor for adverse outcomes during chemoradiotherapy in HNC patients [57]. In patients with sarcopenia, platinum was primarily distributed in fat-free components, and these patients were at a more than three-fold higher chance of developing CDLT than those with normal SMM [2,57]. Jung et al. reported a three-fold increase in the overall risk of recurrence and even death among patients with HNC diagnosed with sarcopenia, underscoring the significance of this condition as a prognostic factor [60]. A systematic review found that low SMM was independently linked to extended breaks in radiotherapy and increased toxicities from chemotherapy in 3461 HNC patients who received curative-intent radiotherapy with or without other treatments [59].

Sarcopenia is also an independent predictor of reduced survival, which provides support for its significance as a prognostic indicator in patients with various malignancies, including ovarian, lung, breast, esophagus, stomach, pancreas, and kidney cancers [61,62,63,64,65,66,67].

Numerous studies showed a lower survival rate in HNC patients with diminished SMM. A systematic review and meta-analysis revealed that lower SMM was linked to shorter disease-free survival (DFS) in surgery-treated patients (HR 2.59, 95% CI: 1.56–4.31) and those receiving radiotherapy for HNC (HR 1.56, 95% CI: 1.24–1.97). A similar relationship was observed for disease-specific survival in surgery-treated patients (HR 2.96, 95% CI: 0.73–11.95) and radiotherapy-treated patients (HR 2.67, 95% CI: 1.51–4.73) [68,69]. Another meta-analysis investigated the relationship between sarcopenia and DFS in 1284 patients undergoing various curative treatments and revealed that sarcopenia predicted DFS in HNC patients (HR 2.00, 95% CI: 1.63–2.45) [46]. Findlay et al. reported data from seven studies (published between January 2004 and May 2020) on 1059 HNC patients treated with radiotherapy and found that a low pre-treatment skeletal muscle index (SMI) was associated with shorter OS (HR 2.07, 95% CI, 1.47–2.92), with similar findings being obtained for a low post-treatment SMI (HR 2.93, 95% CI, 2.00–4.29) [70]. Wong et al. revealed worse OS for HNC patients with low SMM (HR 1.98, 95% CI: 1.64–2.39) [71]. Among HNC patients with sarcopenia treated with surgery or radiotherapy, Takenaka et al. showed that OS was significantly longer in the surgery group (HR 2.50, 95% CI 1.95–3.21) than in the radiotherapy group (HR 1.63, 95% CI 1.40–1.90) [68]. We also reported that low pre-treatment SMM was a strong predictor of poor OS in HNC patients treated chemoradiotherapy [72]. Patients with recurrent oral squamous cell carcinoma exhibited a progressive decline in SMI, which was independently associated with significantly worse OS and recurrence-free survival, even when not reflected in changes in the body mass index (BMI). Therefore, monitoring muscle loss through SMI may provide valuable prognostic information beyond BMI changes [73]. Based on these studies, low SMM is linked to shorter survival in HNC patients across diverse sites, treatment modalities, measurement methods, and time points. The assessment of sarcopenia may contribute to enhanced treatment decision-making.

## 4. Assessment Methods

A more detailed understanding of the risks linked to malnutrition underscores the significance of identifying HNC patients that are the most susceptible to malnutrition before the initiation of treatment. Although various assessment tools for malnutrition have been suggested, a thorough history and physical examination are widely adopted and superior methods for evaluating the nutritional status. Key components of the history to examine encompass recent unintentional weight loss, diminished appetite, and changes in endurance. Malnutrition is characterized by weight loss exceeding 10% of the ideal body weight (IBW) coupled with noticeable muscle wasting. Despite inconsistencies in its definition and assessment, sarcopenia consistently emerges as a significant predictor of adverse outcomes in HNC. It is defined as a low muscle mass, low muscle strength, and poor physical performance [74]. While not a direct measure of muscle mass, low BMI may indicate potential muscle loss [75]. A decrease in mid-arm circumference also suggests muscle wasting [76]. HGS is a common indicator of muscle weakness and may be assessed using a handheld dynamometer. In Asian individuals, the Asian Working Group for Sarcopenia proposed low HGS < 26 kg in men and <18 kg in women [77].

A bioelectrical impedance analysis measures the impedance or resistance of electrical flow through the body and may be used to estimate body composition, including muscle mass, by analyzing how electrical currents pass through different tissues [78]. Dual-energy X-ray absorptiometry scans are primarily used to measure bone density, but also provide information on lean muscle mass [79]. Although formal assessments of sarcopenia, such as muscle strength and function testing, are not standard procedures for oncological prognostication, various methods have been outlined for evaluations of radiological sarcopenia through CT imaging [80]. In the field of oncology, muscle mass has been assessed radiologically using the cross-sectional muscle area (CSMA) at the L3 vertebral body on computed tomography (CT) scans. To account for an individual’s height, the calculation of SMI (cm^2^/m^2^) in CSMA is modified by incorporating the square of a person’s height. Inconsistencies exist in the literature concerning the optimal cut-off values of SMI for defining sarcopenia. Zwart et al. used <43.2 cm^2^/m^2^ as a cut-off value for L3 SMI [9]. Van Rijn-Dekker et al. reported a sex-specific SMI threshold of <42.4 cm^2^/m^2^ in men and <30.6 cm^2^/m^2^ in women [81]. Martin et al. defined sarcopenia using sex-specific cut-offs for L3 SMI (men: 43 cm^2^/m^2^ for BMI < 25 kg/m^2^, 53 cm^2^/m^2^ for BMI > 25 kg/m^2^; women: 41 cm^2^/m^2^) [50]. Inconsistencies in the cut-off values applied to identify sarcopenia in HNC pose challenges to understanding its true impact and making meaningful comparisons between studies. The choice of cut-off values is further complicated by variations in assessment instruments and skeletal muscle measurement sites, making it challenging to establish the most appropriate and clinically relevant criteria for HNC. To address these challenges, the use of a ROC curve analysis is suggested as a preferred method, given its effectiveness, accuracy, and ability to discriminate between patients. A method to examine SMI on a single transverse CT slice at the level of the third cervical vertebra (C3) (paravertebral and sternocleidomastoid muscles) was recently proposed as an alternative to abdominal CT imaging [82], which is not routinely performed on patients with HNC (Figure 2). As cancer-related sarcopenia is inconsistently defined, there is no consensus on cut-off values to guide diagnosis. Although there are multiple thresholds for low SMI, most are not tailored for HNC patients [50,83]. Further research is needed to identify an optimal cut-off value for low SMI in HNC, which is both prognostic and predictive of significant adverse effects. Nevertheless, the measurement of SMI at the C3 level on head and neck CT or MRI imaging has potential as a viable, cost-effective, accurate, meaningful, and rapid biomarker for the identification of sarcopenia. Moreover, the integration of these SMI measurements with fast and dependable assessments of muscle strength, such as HGS, and physical performance will enhance the precision of sarcopenia evaluations in individuals with HNC [84].

## 5. Interventions and Management

Muscle loss of 1–2% per year is expected in healthy adult patients [85]. The prevalence of sarcopenia has been reported to range between 5 and 13% in a population aged 60–70 years and between 11 and 50% in individuals aged ≥ 80 years [86]. Skeletal muscle loss that occurs during cancer and its treatment may involve different mechanisms from those of age-related skeletal muscle loss [87]. Some studies on animal models and clinical trials revealed that sarcopenia and cachexia were mediated by mechanisms including cytokine-mediated inflammation, autophagy, poor nutrition, and physical inactivity [88,89,90]. Therefore, the management of sarcopenia involves a combination of interventions aimed at preventing or slowing down its progression, improving muscle function, and enhancing overall QOL [91]. Due to the prognostic impact of sarcopenia on HNC patients, early identification and appropriate interventions are imperative. Some key aspects of interventions for and the management of patients with sarcopenia are discussed below.

Exercise interventions have been recognized as an essential feature of therapeutic strategies aimed at increasing SMM, knee extension strength, and maximum walking speed [92]. Various types of exercise, including resistance training, aerobic exercise, weight training, whole-body vibration, and balance training, have been shown to increase SMM, muscle strength, and physical performance [93,94,95,96]. However, it is important to note that there are differences in exercise protocols and measurement methods for muscle strength, which lead to high heterogeneity in their outcomes [97,98]. Moreover, HNC patients have poor appetite, nausea, pain, malaise, and neutropenia due to surgery or chemoradiotherapy, and, thus, exercise programs are often unattainable in clinical practice.

While exercise interventions are the cornerstone of treatment for sarcopenia, there are currently no approved pharmacological agents. Steroid hormones, such as testosterone and anabolic steroids, have been reported as potential treatments. Testosterone has shown efficacy at increasing muscle strength and mass [99,100]. Nevertheless, its impact on physical performance remains limited [101]. Other drugs for sarcopenia are currently under development and being tested in clinical trials, including selective androgen receptor modulators [102], estrogen [103], dehydroepiandrosterone [104], insulin-like growth factor-1 [105], growth hormone [106], ghrelin [107], drugs targeting the myostatin and activin receptor pathway [108], vitamin D [109], and angiotensin-converting enzyme inhibitors and angiotensin receptor blockers [110]. Although these agents have been shown to exert positive effects on muscle strength and mass, there are currently no drugs that have achieved clinically relevant improvements in physical performance. The findings of various studies on sarcopenia interventions have been inconsistent, which may be attributed to differences in study designs, the definitions of sarcopenia, and intervention strategies. Kwak et al. discussed pharmacological interventions for the treatment of sarcopenia and emphasized that pharmacological treatments alone were insufficient [111].

Negm et al. reported that the combination of exercise and physical activity with nutritional supplementation was the most effective intervention for sarcopenia; however, more robust clinical trials are needed to improve the quality of evidence [112].

Alcohol and smoking worsen the challenges associated with HNC by depleting essential nutrients. Alcohol and smoking have both been implicated in the development of HNC and may further hinder adequate nutrient intake. The metabolism of alcohol requires specific micronutrients, leading to the additional depletion of nutrient reserves [113]. Moreover, alcohol metabolites may contribute to the growth of cancer and worsen nutritional decline [114]. Smoking also affects nutrient absorption and contributes to oxidative stress and inflammation. Essential nutrients, such as vitamins and minerals, are crucial for supporting immune function, promoting healing, and improving overall treatment outcomes for HNC patients. Therefore, it is crucial for patients with HNC to undergo a nutritional evaluation.

The composition of the gut microbiota has been closely linked to several pathological conditions, such as sarcopenia and cancers [115,116]. The essential interaction between commensal bacteria and the human host is crucial for maintaining homeostasis. Specifically, research has shown that the gut microbiota plays a role in regulating energy extraction from the diet, systemic inflammation, gut barrier function, and insulin sensitivity [117]. These metabolic aspects, which are affected in cancer cachexia, have been identified as key areas affected by the gut microbiota [118]. Various strategies to regulate the gut microbiota have effectively treated sarcopenia in patients with malignancies. Supplementation with Lactobacillus reduced muscle atrophy in mice, and a symbiotic intervention involving insulin-type fructans and Lactobacillus reuteri attenuated muscle wasting and prolonged survival in tumor-bearing mice [119,120]. A multifaceted approach, combining enhanced nutrition with modulations of the gut microbiota, may be more effective due to the complex and multifactorial nature of sarcopenia.

## 6. Future Directions

Various body features, including skeletal muscle, adipose tissue, and BMI, whether assessed in combination with the nutritional status, are of predictive value in HNC patients receiving chemoradiotherapy or undergoing surgery. These factors need to be considered in future research on sarcopenia. The routine use of a body composition analysis through CT or MRI imaging may be a valuable addition to the management of HNC patients and clinical decision-making. However, a consensus on the assessment and definition of sarcopenia is needed to validate these findings and support the integration of a body composition assessment as a clinically meaningful prognostic tool. Further studies are required to identify effective preoperative exercise and nutrition programs targeting low SMM for better treatment outcomes and longer survival.

## 7. Conclusions

HNCs are a significant global health concern, and sarcopenia is a prevalent issue among patients with these cancers. The relationship between HNC and sarcopenia is intricate, involving factors such as shared etiological factors, cancer treatments, and inflammation. Sarcopenia has profound clinical implications, affecting QOL, surgical outcomes, chemotherapy tolerance, and overall survival. Assessment methods for sarcopenia vary, and its management necessitates a multidisciplinary approach, including exercise, nutritional support, and potential pharmacological interventions; however, there are currently no approved drugs for the treatment of sarcopenia. Early identification and interventions are crucial for improving the prognosis and QOL of HNC patients with sarcopenia.

## Figures and Tables

**Figure 1 cancers-16-03460-f001:**
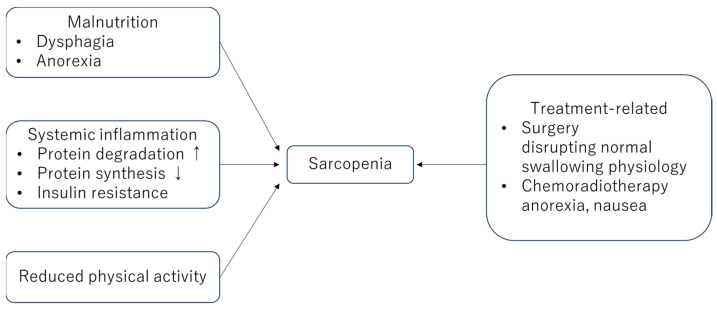
Mechanisms underlying HNC-related sarcopenia. Sarcopenia is mediated by multiple mechanisms, including malnutrition, systemic inflammation, reduced physical activity, and treatment.

**Figure 2 cancers-16-03460-f002:**
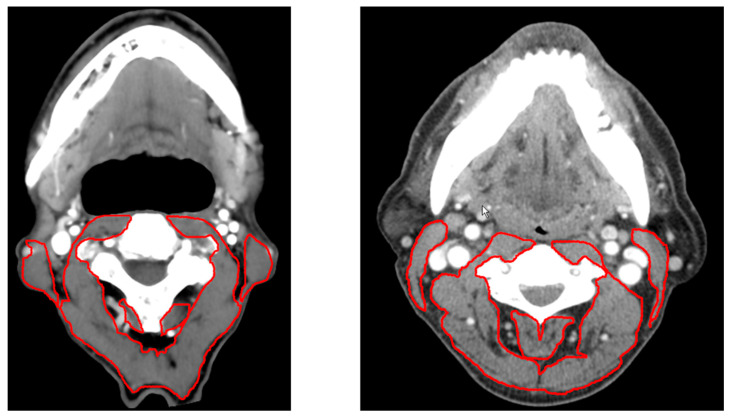
Representative cases of sarcopenia (**left side**) and non-sarcopenia (**right side**). The part outlined in red indicates the paravertebral muscles (PVMs) and sternocleidomastoid muscles (SCMs).

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
