# Peer review of "Head and Neck Cancer and Sarcopenia: An Integrative Clinical and Functional Review"

_cancers, 2024, doi:10.3390/cancers16203460_

Round 1

Reviewer 1 Report

Comments and Suggestions for Authors

The authors have reviewed the topic of sarcopenia in patients with cancer of the head and neck. Many articles are cited to discuss the definition and assessment of sarcopenia, as well as interventions.  Sarcopenia is an important condition in patients with head and neck cancer, so this review will be of interest to readers of Cancers.

Several minor points need to be corrected.

 1. The sentence referring to the macrobiome in the abstract seems somewhat abrupt. The microbiome is just only one of the topics which was introduced in the main text. If the microbiome is to be presented in the abstract, more convincing evidence needs to be presented in detail.

2. p5, l189 “...low SMM is linked with disease-free survival”: this text should describe how SMM is associated with DFS. For example, SHORTER disease-free survival.

3. p5, l201 “Among HNC patients treated with surgery and radiotherapy...”: the cited review is about head and neck cancer patients with sarcopenia. Therefore, the adjective “sarcopenia” should also be added to somewhere in this sentence.

Author Response

  1. The sentence referring to the macrobiome in the abstract seems somewhat abrupt. The microbiome is just only one of the topics which was introduced in the main text. If the microbiome is to be presented in the abstract, more convincing evidence needs to be presented in detail.

    We have decided to remove the mention of the microbiome from the abstract and discuss it in detail in the main text.

  2. p5, l189 “...low SMM is linked with disease-free survival”: this text should describe how SMM is associated with DFS. For example, SHORTER disease-free survival.

    As the reviewers pointed out, the text has been revised. A systematic review and meta-analysis revealed that lower SMM was linked to shorter disease-free survival (DFS) in surgery-treated patients (HR 2.59, 95% CI: 1.56–4.31) and those receiving radiotherapy for HNC (HR 1.56, 95% CI: 1.24–1.97). P5 l188
  3. p5, l201 “Among HNC patients treated with surgery and radiotherapy...”: the cited review is about head and neck cancer patients with sarcopenia. Therefore, the adjective “sarcopenia” should also be added to somewhere in this sentence.

    As the reviewers pointed out, the text has been revised. Among HNC patients with sarcopenia treated with surgery or radiotherapy, Takenaka et al. showed that OS was significantly longer in the surgery group (HR 2.50, 95% CI 1.95–3.21) than in the radiotherapy group (HR 1.63, 95% CI 1.40–1.90). P5 l200

Reviewer 2 Report

Comments and Suggestions for Authors

This review paper is based on previously reported research papers and comprehensively describes the impact of sarcopenia on head and neck cancer patients from various perspectives based on scientific evidence, and I felt that it is very useful for clinical application to head and neck cancer patients. However, there are some points that require additional explanation, so please revise these.

Line 71-72 “Sarcopenia may cause patients to stop or interrupt radiation therapy and/or chemotherapy.”

The sentences before and after this statement do not explain why sarcopenia may cause patients to stop radiation therapy or chemotherapy. The reason is stated from line 164 onwards, but it is quite far from this sentence. It should be revised to make it easier for readers to understand.

Figure 2 What does the red box indicate? Please explain in the figure caption.

Line 309-310 “Alcohol and smoking worsen the challenges associated with HNC by depleting essential nutrients.”

Please add an explanation for this statement. It is stated later that alcohol depletes nutrients, but it does not state that smoking has a similar effect. Also, it is necessary to explain how essential nutrients affect the problems associated with HNC.

Author Response

  1. The sentences before and after this statement do not explain why sarcopenia may cause patients to stop radiation therapy or chemotherapy. The reason is stated from line 164 onwards, but it is quite far from this sentence. It should be revised to make it easier for readers to understand.

    As the reviewer pointed out, the discussion on between the continuity of sarcopenia and CRT will be consolidated in the later section. The mentioned sentence will be removed
  2. Figure 2 What does the red box indicate? Please explain in the figure caption.

    We added the explanation about red line in Figure 2. “The part outlined in red indicates the paravertebral muscles (PVM) and sternocleidomastoid muscles (SCM).”

  3. It is stated later that alcohol depletes nutrients, but it does not state that smoking has a similar effect. Also, it is necessary to explain how essential nutrients affect the problems associated with HNC.

    As the reviewer pointed out, we added the following explanation. “Smoking also affects nutrient absorption and contributes to oxidative stress and inflammation. Essential nutrients, such as vitamins and minerals, are crucial for supporting immune function, promoting healing, and improving overall treatment outcomes for HNC patients.” P7 l315